# Palm oil: Understanding barriers to sustainable consumption

**Cassandra Shruti Sundaraja**[1], **Donald W. Hine**[2], **Amy D. Lykins**[1]*

**1** University of New England, Armidale, NSW, Australia, **2** University of Canterbury, Upper Riccarton, Christchurch, New Zealand

* alykins@une.edu.au

**Data Availability Statement:** The data files (data set and SPSS output files) associated with this project are located in a public repository and can be found at https://hdl.handle.net/1959.11/29488 (doi: 10.25952/5f71639941626).

## Abstract

Palm oil is relatively inexpensive, versatile, and popular, generating great economic value for Southeast Asian countries. However, the growing demand for palm oil is leading to deforestation and biodiversity loss. The current study is the first to employ a capability-opportunity-motivation (COM-B) framework in green consumerism, to determine which capability, opportunity, and motivation factors strongly predict the intentional purchasing of sustainable palm oil products by Australian consumers (N = 781). Exploratory factor analysis revealed four main types of predictors of SPO purchasing–*Pro-Green Consumption Attitudes*, *Demotivating Beliefs*, *Knowledge and Awareness*, and *Perceived Product Availability*. Multiple regression revealed that these four factors explained 50% of the variability in SPO purchasing behaviour, out of which *Knowledge and Awareness* accounted for 18% of the unique variance. *Perceived Product Availability* and *Pro-Green Consumption Attitudes* were also significant predictors but accounted for only 2% and 1% of unique variance, respectively. These results provide a valuable foundation for designing behaviour change interventions to increase consumer demand for sustainable palm oil products.

## Introduction

The focus on global climate change intensified in 2019 with the unprecedented ferocity of the bushfire season in Australia, dubbed as the "Black Summer", owing to high temperatures and the pre-existing drought [1, 2]. The ongoing fire season in western United States is anticipated to mirror the conditions seen in Australia, exacerbated by increased temperatures and a slow recovery from drought [3]. Additionally, the Atlantic hurricane season has seen devastating storms hit parts of the United States of America, which could be due to warmer ocean temperatures that fuel cyclones [4]. The year 2019 also saw an increase in the worldwide destruction of tropical rainforests, despite commitments made by companies and governments to decrease deforestation by 2020 [5]. While children–inspired by Greta Thunberg, a teenage environmental activist–left schools to protest climate-related inaction [6], tropical forests in the Amazon region [7] and in South East Asia [8, 9] continue to burn for agricultural purposes. These main belts of tropical rainforests are essential for regulating global and local temperatures [10], the destruction of which has huge implications for climate change.

**Funding:** The authors received no specific funding for this work.

**Competing interests:** The authors have declared that no competing interests exist.

The idea that population growth and rates of human consumption cannot be sustained and are causing irreparable damage to the environment has been around since the late 1960's and early 1970s [11–13]. There have been warnings about planetary boundaries and thresholds being crossed with respect to climate change, land use and biodiversity loss, which threaten a safe operating space for humanity [14]. The exponential growth of industrialization and food production has been feared to lead to an impending collapse [13, 15]. Almost 50 years later, the message continues to be the same–our patterns of consuming food, water, energy and other natural resources need to change in order to effectively address environmental challenges [16]. At the same time, there is concern about economic development and progress, and ensuring that needs are met in under-developed and developing countries [11, 16]. With growing attention on the impact of unchecked agriculture on climate change [17], this conflict between economic growth and environmental protection and preservation is well-illustrated with the issue of palm oil.

The global rising demand for edible oils, including palm oil, is fuelled by increasing per capita incomes, growing consumerism, and changing lifestyles [18, 19]. With about 80 to 90 per cent of palm oil produced for human food consumption, and the remaining 10 per cent consumed by various industries, such as biodiesel, cosmetics and pharmaceuticals [20], the global palm oil market was estimated to be 74.6 million tonnes in 2019 [21]. This is further expected to increase to 111.3 million tonnes by the year 2025 [22]. Grown predominantly in Southeast Asia, it is unsurprisingly viewed by many farmers there as a miracle crop due to its high yield, versatility, and relatively low production costs [23]. With the global market for palm oil growing rapidly, Indonesia and Malaysia rely on oil palm plantations for economic development and stability [18]. Over 40% of oil palm plantations are run as family farms [18]; as such, the livelihoods of these rural farmers rely on palm oil [24].

In order to keep up with the increasing demands for palm oil, however, tropical rainforests are cleared on a large-scale to make room for plantations, which has adversely impacted the biodiversity in Southeast Asia [8, 18, 25–28]. Several species, including the Bornean orangutan and the Sumatran tiger are now critically endangered. Moreover, these rainforests were often cleared using the popular slash-and-burn technique, releasing enormous amounts of greenhouse gases like carbon dioxide, as the peatlands under the forests are set fire to and drained [29–31]. This burning literally turns the sky red in parts of Indonesia [9], and the resultant haze (dubbed the 'Southeast Asian Haze) causes acute respiratory illnesses and is estimated to have claimed over a 100,000 lives [8, 32, 33]. More recent satellite monitoring has identified that non-forested land may be burned more often than forested land, although the use of fire in deforestation continues to be used by both independent farmers and large companies [34].

A boycott of palm oil would not only be impractical [35] and hurt the interests of rural farmers, but would also merely displace the deforestation as palm oil has the greatest land-use efficiency among all other oilseed crops [18]. Should another oil be used to replace palm oil, it would only result in increased crop-related deforestation elsewhere [18]. Technological solutions, including creating synthetic oils to replace palm oil, are currently very expensive [23]. As a consequence, Parsons, Raikova and Chuck [23] have proposed that promoting more "sustainable" practices in the palm oil industry may be the most feasible short- to medium-term for minimising its environmental impacts. The idea of sustainability initially took shape with the Brundtland Report of 1987, which stated that "Sustainable development is development that meets the needs of the present without compromising the ability of future generations to meet their own needs" [36, p. 41]. It is generally agreed that dimensions of social, economic, and environmental sustainability are all relevant [37, 38], although the relative importance of each dimension is subject to vigorous debate [38].

The Roundtable on Sustainable Palm Oil (RSPO) was set up in 2004 as a non-governmental body to regulate and certify palm oil based on certain sustainability criteria [39], and represents one possible pathway to increased sustainability. However, there continues to be scepticism and suspicion around the term sustainability, particularly because it is difficult to quantify and measure [38]. Research into the efficacy of the RSPO in performing its role has received mixed results. One study that reported lower rates of deforestation in RSPO-certified plantations also revealed that there were fewer intact forests there prior to certification [40]. In alignment with the dimensions of sustainability, there were no significant differences on environmental, social, nor economic metrics between RSPO-certified and non-certified plantations [41]. RSPO certifications have been accused of "greenwashing" palm oil by labelling it as sustainable, even when the sustainability criteria have not been entirely met [42–45]. Even after a recent revision of the RSPO's Principles and Criteria [37], the 2019 case studies continue to highlight its shortcomings [44]. Although advocating for the purchase of sustainable palm oil products is far from a perfect solution, many argue that it may be the best solution currently available [18, 35, 45].

A primary goal of the present study was to identify and understand which factors promote or discourage the purchase of sustainable palm oil. The literature on drivers and barriers of green, sustainable, and ethical consumption is diverse and growing [e.g. 46–52]. In their systematic review, Joshi and Rahman [53] examined 53 empirical articles (published between 2000 and 2014) on green purchasing behaviour and intention. Across habitual and one-time purchases (e.g., plastic-free products and an energy-efficient washing machine respectively), they identified a list of individual (emotions, habits, perceived consumer efficacy, perceived behavioural control, values, personal norms, trust in green products, and knowledge about environmental issues) and situational factors (price, product availability, social norms, product attributes and/or quality, store-related attributes, brand image, eco-labelling and certifications) that were important [53].

Quantitative studies in green consumerism tend to focus on a relatively narrow range of sustainable consumption predictors like willingness to pay and personal norms [51], or self-efficacy [54], which risks ignoring other factors that may influence consumer decision-making and behaviour. Further, studies that investigate green behaviour in general may overlook unique barriers to specific green consumer behaviours (e.g., the consumption of SPO).

Few studies have explored specific barriers and drivers to green purchasing behaviour beyond motivation-related factors [55–57], and we found only one that focussed directly on purchasing SPO products [56]. This Swedish ethnographic study stressed the importance of limited available information and numerous choices, as well as the difficulties consumers face with respect to the complexity of the palm oil issue combined with limited time to research the environmental issue [56].

Although there has been considerable research on green consumerism, the field lacks an integrative framework for organising all the potential drivers and barriers to the purchasing of green products. The behaviour change wheel (Fig 1) was developed by Michie, Van Stralen and West Michie, Van Stralen [58], initially for applications in health psychology. It is a comprehensive framework for identifying the causes of behaviour, and linking these causes to appropriate behaviour change and policy initiatives [58]. The inner circle of the behaviour change wheel contains Capability-Opportunity-Motivation (COM-B) factors that influence behaviour, and interact in varying proportions to produce or prevent specific behaviours [58]. These components can further be sub-divided as follows:

1. Capability: Physical (such as having the physical skills necessary to perform the behaviour) and/or psychological (which includes knowledge, cognitive skills, and the capacity to

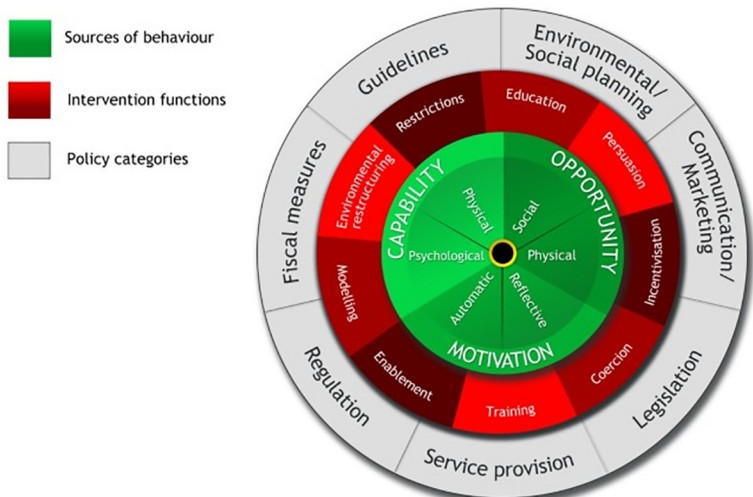

**Fig 1. The behaviour change wheel.** Reprinted from "The behaviour change wheel: A new method for characterising and designing behaviour change interventions," by S. Michie, M. M. van Stralen and R. West, 2011, Implementation Science, 6(42), p. 7. Copyright [2011] by Michie et al., licensee BioMed Central Ltd. Used under Creative Commons Attribution License: http://creativecommons.org/licenses/by/2.0.

regulate behaviour). In green consumerism, capability can refer to knowledge on the environmental and/or social issue, and affordability or one's financial status [53].

2. Opportunity: Physical (referring to environmental resources and contexts) and/or social (the influence of friends, family, and/or society). Opportunities to engage in green consumer behaviour could be facilitated by visible branding on sustainable products, large eco-labelling, availability of green products, social norms, and the like [53].

3. Motivation: Automatic (under which emotions and rewards fall) and/or reflective (components of evaluation for decision-making including intentions, goals, self-efficacy, etc.). A consumer's motivation to purchase green products can stem from empathy, guilt, concern for the environment, perceived consumer efficacy, personal values, among others [53].

The COM-B model of understanding behaviour was proposed as a behaviour "system" where all three elements interact with one another to generate behaviour, which in turn affects these elements of capability, opportunity, and motivation [58]. As there is no hierarchical structure among the COM components, all can have equally important influences on behaviour [58]. However, in order to target a specific behaviour with an intervention, it is essential to identify where the main barriers lie–that is, within capability, opportunity or motivation [58].

COM-B and the behaviour change wheel have mostly found their use in the field of health, to promote lifestyle changes like exercising more, reducing or ceasing smoking, and the like [59–63]. However, research in the pro-environmental space has rarely used the COM-B model and the current research is the first to explore its application in understanding and promoting green or sustainable consumption (which can be comparable to the health behaviours addressed by the behaviour change wheel in literature). Given that there are a variety of factors across the COM-B that could potentially influence the purchase of sustainable palm oil, it is essential to identify which specific barriers would need to be addressed in a targeted intervention in order to ensure optimal allocation of resources.

This paper will describe two studies–a preliminary qualitative research study, Study 1 [64], designed to identify the main drivers and barriers of SPO purchasing behaviour and then organise them according to the COM-B framework. Following this, in a quantitative research study, Study 2, we developed a new measure based on the identified drivers and barriers, in order to assess capability, opportunity and motivational factors relevant to the purchase of products containing SPO. Further, the factor structure of this measure and its capacity to predict self-reported SPO purchasing behaviour were also assessed.

## Materials and method

### Study 1 [64]

**Participants.** In this research study, the first author interviewed 13 Australian consumers (aged between 24 to 73 years; median age = 30 years) with varying levels of engagement on sustainable palm oil purchasing behaviour. Some participants were recruited via recruitment posters put up in areas of community gathering (e.g., churches, the university campus, and sports centres) in a regional town in New South Wales, Australia, as well as on social media. In addition to this, research participation credit was offered to first-year psychology students at a regional university known to have a diverse online student body from all over Australia.

**Procedure.** Participants were interviewed face-to-face, or using a video-conferencing platform (i.e., Zoom), based on an interview guide. The interview questions explored various barriers and drivers relating to capability, opportunity, and opportunity [58], that play a role in decisions around SPO-related consumer behaviour. The length of interviews ranged from 19 to 38 minutes.

**Data analysis.** These qualitative interviews were subjected to framework analysis [65] based on COM-B. Framework analysis is a means of qualitative data analysis that commences with identifying a thematic framework and then coding the collected data based on this framework [65]. Once this is done, summaries of data are arranged under each thematic framework (known as charting), after which they are interpreted by identifying associations between themes, as well as making within-case and between-case comparisons, thereby explaining the findings [65].

**Ethics.** The study was approved by the Human Research Ethics Committee of the University of New England, Australia (Approval No. HE19-032) and was conducted in compliance with the recommended research ethics procedure. All participants provided written informed consent.

**Results.** A summary of these charted findings is listed in Table 1 and detailed information on data collection, analysis and results is to be published elsewhere [64].

**Table 1. COM-B factors influencing the purchase of SPO.**

| Capability | Opportunity | Motivation |
|---|---|---|
| Knowledge about the issue | SPO product availability | Empathy, compassion and love for the environment |
| Critical thinking | Legible, clear product labels | Guilt |
| Capacity to afford 'green' products | Visibility | Perceived consumer efficacy |
| Time | Social norms | Shopping habits |
| Energy | | Values / Moral compass |
| | | Health benefits |

The next study builds upon this qualitative study and aims to apply the COM-B framework to further understand which specific factors best predict engagement in sustainable palm oil-related consumer behaviour.

## Study 2

**Participants.**   The sample consisted of 781 adult participants from Australia who self-reported as the primary grocery/supermarket shopper in their household. A target sample size was determined by a power analysis. A study that studied drivers and barriers associated with the adoption of low emission agricultural practices among farmers, and thus had some parallels to the proposed study, found small effect sizes for their psychological variables [squared semi-partial correlations ranging from $< .01$ to $.10$; 66]. These variables reflected some aspects of the COM-B model. Based on this research, a small effect size of $f^2 = .02$ [67, 68] is predicted. Assuming a target power of .90, 10 predictors and an $\alpha$ level of .01, a power analysis using G*Power [69] suggested a minimum sample size of 748 participants.

All participants were sourced using a survey panel from Qualtrics$^{TM}$, an online database and survey administrator [70 Provo, UT]. Qualtrics' samples come from market research panels and respondents are invited to participate in various ways, often via an email invitation. Other means of invitations include displaying surveys when potential respondents sign into a panel portal, or the use of in-app and SMS notifications. To avoid a self-selection bias, survey invitations do not include specific details about the contents of the survey and are instead kept very general (i.e., information that the survey is for research purposes only, how long the survey is expected to take, and what incentives are available).

The sample was representative across adult age groups, mapped on the national representative census data. Participants were predominantly female (498 women, 283 men) likely due to the screening procedures utilized to survey the member of the household that primarily does the grocery shopping. Although trends are changing, grocery shopping has been traditionally viewed as a female-centric gendered role, and recent surveys continue to indicate that women are more likely than men to perform routine grocery purchases [71, 72]. Respondents had a mean age of 46 years ($SD = 17.26$, range = 18 to 83 years), and they reported having completed an average of 13 years of formal education. Almost three-quarters of the participants (73%) stated that they lived in an urban area (big city or large town) and 35% indicated an annual household income of less than 50,000AUD [the average Australian household income for 2017 to 2018 was approximately 55,000AUD; 73].

**Measures.**   The measures used in this study included an indication of the existing frequency of sustainable palm oil-related consumer behaviour, and a specially designed palm oil-related COM-B survey, derived from the preliminary study described above. In addition to this, established measures of connectedness to nature, willingness to sacrifice for the environment and values were also administered, as these have been found to be significant predictors of pro-environmental behaviour in the past [74–78]. This combination of a new measure and existing measures was used to see if the palm oil-related COM-B survey better explained sustainable palm oil consumer behaviour than previously used measures in this field.

*Basic socio-demographic details*. Participants were asked relevant, non-identifying questions about age, sex, years of education, residence (urban or rural/regional), and annual household income.

*Frequency of sustainable palm oil-related consumer behaviour*. Four items on specific behaviour pertaining to the purchasing of products containing sustainable palm oil were included ("*At a supermarket/food store, how often have you intentionally looked for products that contain sustainable palm oil?*", "*How often have you intentionally purchased a product because it*

*contained sustainable palm oil*?", "*How often have you avoided buying a product when you real-ized that it contained palm oil that was not sustainable*?", and "*How often have you returned a product to a store because it contained unsustainable palm oil*?"). Participants were asked to provide an indication of frequency of each behaviour on an 11-point scale, where 0 = "never" and 10 = "10 or more times." These four items were summed into a scale, which yielded a Cronbach's α of .85. Cronbach's α is a measure of reliability and determines the extent to which all items in a scale measure the same construct [79]. Also referred to as internal consis-tency, it refers to the inter-relatedness of the items within the test [79]. For the current mea-sure, Cronbach's α increased to .88 when the last item "*How often have you returned a product to a store because it contained unsustainable palm oil*?" was deleted, indicating that the scale's internal consistency improved without this item. As this last item also had a highly positively skewed distribution, with 81.4% of the sample reporting that they had never performed this behaviour, only the first three items were totalled and retained for subsequent analysis.

*Palm oil COM-B survey*. Based on interviews conducted with Australian consumers (Study 1), several barriers and drivers of purchasing products with sustainable palm oil were identi-fied (Table 1). These were then incorporated into a survey of 53 items that included items about capability (for example, knowledge about the environmental effects of the palm oil industry, affordability, limited resources of time and energy), opportunity (e.g., availability, clear and legible labelling of ingredients, discussions with friends/family, and social norms), and motivation (e.g., personal norms, habits, concern for future generations, satisfaction/ pride, perceived consumer efficacy, concern for the environment, and trust in the quality of green products). Initially, all items were worded specific to sustainable palm oil. However, when these items were pilot-tested, it was discovered that if participants were unaware of the palm oil crisis and the option of sustainable palm oil, it was extremely difficult for them to answer the subsequent questions around COM-B. Therefore, a decision was made to retain the items assessing knowledge specific to sustainable palm oil, and to alter the other items to pertain to "green" or "sustainable" products in general. All items were rated on a 5-point scale of 1 = "not at all like me," to 5 = "just like me." These items were subjected to an exploratory factor analysis, the details of which are summarised in the Results section. Cronbach's α for the scaled factors ranged from .79 to .95.

*Connectedness with Nature Scale [CNS; 74]*. The CNS is a widely used scale in environmen-tal research that assesses an individual's connection with nature with respect to their cogni-tions (i.e., beliefs). Fourteen items consisting of statements written in the first person are each rated on a 5-point scale, where 1 = "strongly disagree" and 5 = "strongly agree," which were averaged to provide a CNS score. This scale is reported to have high internal consistency (Cronbach's α = .84), high test-retest reliability (r = .78, p < .001), and has demonstrated con-current, convergent and discriminant validity [74]. In the current study, the CNS had a com-parable internal consistency with α = .86.

*Willingness to Sacrifice for the environment [WTS; 75]*. The WTS is a 5-item measure of an individual's willingness to sacrifice their own needs for the sake of the environment. Each item comprised a 9-point Likert scale, ranging from 0 = "do not agree at all," to 9 = "agree completely." All five items were averaged to produce a WTS score. Previous research indicates that the WTS has high internal consistency [Cronbach's α = .88; 75], and correlates strongly with other measures of pro-environmental attitudes, including the New Ecological Paradigm [80], Inclusion of Nature in the Self [81], and the CNS [74], with correlations ranging from r = .35 to .60 [p < .001; 75]. In the current study, the internal consistency of WTS was very high at α = .96.

*Adaptation of Schwartz's value scale [82]*. Based on Schwartz's original 56 values [82], de Groot and Steg [76] developed a taxonomy of those that are most relevant in the context of

pro-environmental behaviour. These are values related to the self-enhancement versus self-transcendent dimensions, and consist of egoistic, altruistic and biospheric value orientations [76]. Participants indicated to what extent each of the 13 values or items is important "as a guiding principle in their lives" on a 9-point scale (–1 = "opposed to my values," 0 = "not important," to 7 = "extremely important"). Following the original instructions [76, 82], participants were asked to vary scores as much as possible and to rate no more than two values as extremely important. Items under each scale (5 items for egoistic, 4 items each for altruistic and biospheric value orientations) were averaged to obtain the respective scale scores. As a measure of internal consistency, Cronbach's α in a European sample was found to be .74 for the egoistic, .73 for the altruistic, and .86 for the biospheric value orientations [83], while in the current study, the internal reliabilities obtained were α = .84, α = .81, and α = .90, respectively.

**Procedure.** Participants were recruited via a Qualtrics^TM online panel [70 Provo, UT]. They were provided with an information sheet, after which they could provide consent and then complete the survey. Those who were below 18 years of age, and those who reported that someone else (apart from the participant themselves) primarily did the grocery and/or super-market shopping for the household, were screened out of the survey. In the survey, all questions were mandatory (although participants could select "Rather not say" for the question about annual household income). Except for the socio-demographic and frequency of sustainable palm oil-related consumer behaviour (which were presented at the start of the survey, so that participants were not primed by their responses on measures of attitudes and values towards the environment and 'green' products), all other measures were presented in a random order, and items within each measure were also randomized. As attention-checks, two instructed-response items (e.g., "In order to check for attention, please answer '1' for this question") were incorporated into the palm oil-related COM-B survey. These instructed-response items have been known to be useful in screening out careless responders, while still protecting the validity of the scale [84, 85]. Qualtrics^TM screened out participants who failed either one of the attention-checks or provided only partial responses. Additionally, a speed-checker was incorporated into the online survey. Participants whose response times were approximately below one-half of the average time taken to complete the survey in a soft launch (i.e., less than or equal to six minutes) were automatically screened out as well to ensure good data quality. As an outcome of the attention-check and speed-check screening procedures, 15 participants were screened out from an original dataset of 796, resulting in the current sample of 781 participants.

**Data analysis.** Exploratory factor analysis and linear regression were run using IBM SPSS Statistics 26 (IBM corp., Armonk, NY, USA).

**Ethics.** The study was approved by the Human Research Ethics Committee of the University of New England (Approval No. HE19-223) and was conducted in compliance with the recommended research ethics procedure.

**Results.** Participants reported low rates of engaging in sustainable palm oil-related consumer behaviour, with 40% of the sample reporting never having engaged in any sustainable palm oil-related action over the past year, including checking labels for sustainable palm oil products, avoiding products containing unsustainable palm oil, or purchasing products containing sustainable palm oil.

**Exploratory factor analysis.** The Palm oil COM-B survey consisted of 53 items, some of which were phrased as drivers and others as barriers. All the driver items were reverse-scored, so that in the analyses, all of the items reflected barriers to purchasing sustainable palm oil. These items were then subjected to exploratory factor analysis, using the maximum likelihood extraction method, to determine the underlying structure of the variables and to reduce the data into more manageable units. The Kaiser-Meyer-Olkin measure of sampling adequacy was

.95, and Bartlett's test of sphericity had a *p*-value of less than .001, which indicated that there were strong linear relationships within the data set. The number of factors to retain was decided based on Cattell's [86] scree plot and parallel analysis [87, 88]. The scree plot distinctly indicated that three factors should be retained, whereas raw parallel analysis proposed six factors. All possible solutions for three to six factors were run using the direct oblimin rotation with Δ set to 0. A four-factor solution (*Pro-Green Consumption Attitudes*, *Knowledge and Awareness*, *Demotivating Beliefs and Perceived Product Availability*) was the most interpretable and accounted for 45% of the overall response variance. The five and six factor solutions had a high number of cross-loading items across two or more factors and there appeared to be conceptual overlap between the additional factors generated. This four-factor structure has been replicated on an independent data set in research, currently under review elsewhere [89].

Items that loaded greater than .45 on one factor with a cross-loadings of .35 or less were used to define each factor. Scale scores for each of the retained factors were computed by taking the average across the selected items for each factor, and were used for subsequent analysis. Table 2 lists the items included under each factor, the mean and standard deviation (prior to reverse-scoring), and the internal reliability score for each factor, after all driver items were reverse-scored to represent barriers.

**Multiple regression.** A multiple regression analysis was conducted to predict the frequency of SPO-related consumer behaviour, based on the COM-B factors (Table 2), as well as the measures on CNS, WTS and values. Together, the four COM-B derived factors explained 50% of the variance in the frequency of SPO-related consumer behaviour. A summary of the regression analysis is presented in Table 3. Further, including the CNS, WTS and Value Orientations (Egoistic, Altruistic and Biospheric) in the regression model only added an additional 1% to the amount of variance predicted in the outcome variable ($R = .72$, $R^2 = .52$, adjusted $R^2 = .51$, $p < .001$). Examining the beta-coefficients of the COM-B factors, it is apparent that *Pro-Green Consumption Attitudes*, *Knowledge and Awareness*, as well as *Perceived Product Availability* explained significant amounts of unique variance in the frequency of SPO-related consumer behaviour, with *Knowledge and Awareness* explaining the most amount of unique variance (18%). Therefore, these results indicate that reducing barriers associated with *Knowledge and Awareness*, *Perceived Product Availability*, and *Pro-Green Consumption Attitudes* are more likely to positively influence consumers to engage in more SPO purchasing behaviour.

## Discussion

This research was the first to adopt a systemic COM-B framework to identify specific barriers to the sustainable consumption of palm oil. The results revealed that barriers related to a lack of knowledge and awareness about the issue reduced perceived product availability, and fewer pro-green consumption attitudes best predicted engagement in SPO purchasing behaviour, which directly mirror Capability, Opportunity and Motivation (COM-B) respectively. These results, therefore, demonstrate the successful application of COM-B in the consumerism space, and illustrate its utility in understanding the nature of specific barriers and drivers to consumer PEB. While the COM-B model has been successfully implemented in understanding and modifying health-related behaviour [59–63], its evidence-base in the pro-environmental space is in the early stages [90]. Further, our findings reinforce the idea that research focusing exclusively on internal motivation-related factors are likely overlooking essential impacts of capability and opportunity on the target behaviour [56, 91]. The current research reveals that factors relating to all three categories of the COM-B are implicated in predicting the frequency of SPO-related consumer behaviour.

**Table 2. Exploratory factor analysis pattern matrix loadings (N = 781).**

| Palm oil COM-B items | M (SD)# | 1 | 2 | 3 | 4 |
|---|---|---|---|---|---|
| | | \multicolumn{4}{} Factor loading | | | |
| **Factor 1 –Pro-Green Consumption Attitudes (Motivation; α = .95)** | | | | | |
| I feel satisfied when/if I buy sustainable products. (R) | 3.54 (1.18) | .81 | | | |
| I feel proud when/if I buy green products. (R) | 3.39 (1.23) | .81 | | | |
| I feel responsible for protecting the environment by purchasing green products. (R) | 3.36 (1.20) | .78 | | | |
| I am concerned about preserving our planet for future generations. (R) | 3.88 (1.10) | .78 | | | |
| I see myself as a person who cares about the environment. (R) | 3.92 (0.99) | .75 | | | |
| I worry about the state of the planet, what we will leave behind for my children, grandchildren and great-grandchildren. (R) | 3.78 (1.21) | .75 | | | |
| I can make a difference to the environment if I purchase sustainably. (R) | 3.54 (1.12) | .74 | | | |
| I feel a strong personal responsibility to buy green products. (R) | 3.13 (1.24) | .70 | | | |
| I feel guilty when I purchase products that are bad for the environment. (R) | 3.27 (1.30) | .68 | | | |
| I believe that consuming green products benefits my health. (R) | 3.43 (1.21) | .67 | | | |
| It is important for me to purchase sustainable products. (R) | 3.29 (1.18) | .65 | | | |
| I empathize with animals that are affected by human farming activity. (R) | 3.94 (1.14) | .61 | | | |
| I see myself as a person who cares about animal life. (R) | 4.15 (0.93 | .58 | | | |
| It is worth paying a higher price for green products. (R) | 2.95 (1.26) | .55 | | | |
| I generally take animal welfare into account while shopping. (R) | 3.38 (1.27) | .54 | | | |
| I can influence others around me by encouraging them to shop sustainably. (R) | 2.92 (1.19) | .52 | | | .31 |
| **Factor 2—Knowledge & Awareness (Capability; α = .91)** | | | | | |
| I am aware of the effects of palm oil production on forests. (R) | 3.17 (1.47) | | .93 | | |
| I am aware of the effects of palm oil production on certain animal species. (R) | 3.09 (1.4) | | .90 | | |
| I am aware of how palm oil production impacts locals in Southeast Asia (farmers, plantation workers, etc.). (R) | 2.92 (1.42) | | .86 | | |
| I know the difference between sustainable palm oil and ordinary palm oil. (R) | 2.55 (1.39) | | .70 | | |
| I know which products contain palm oil. (R) | 2.40 (1.27) | | .63 | | .32 |
| I have heard about sustainable palm oil. (R) | 2.89 (1.45) | | .63 | | |
| **Factor 3—Demotivating Beliefs (Motivation; α = .79)** | | | | | |
| It is exhausting to change my purchasing behaviour for environmental reasons. | 2.59 (1.14) | | | .61 | |
| I think that sustainable products are less tasty. | 2.56 (1.15) | | | .57 | |

(*Continued*)

**Table 2.** (Continued)

| Palm oil COM-B items | M (SD)# | Factor loading | | | |
|---|---|---|---|---|---|
| | | 1 | 2 | 3 | 4 |
| It is hard to give up products I like, even if I know they are not good for the environment. | 3.17 (1.11) | | | .55 | |
| I think that green products are often poorer in quality. | 2.43 (1.16) | .34 | | .51 | |
| It is inconvenient for me to purchase green products. | 2.54 (1.11) | | | .50 | |
| I feel that green products offer poor value for money. | 2.93 (1.23) | .32 | | .48 | |
| Despite my good intentions, I often forget to purchase green products. | 3.12 (1.11) | | | .47 | |
| I am often rushed for time when I go shopping for food and household supplies. | 2.85 (1.29) | | | .47 | |
| Getting the shopping done quickly is my top priority. | 3.28 (1.28) | | | .46 | |
| **Factor 4—Perceived Product Availability (Opportunity; α = .84)** | | | | | |
| In a supermarket, I know where exactly (e.g. in which aisle) I can find green products. (R) | 2.71 (1.28) | | | | .64 |
| I know where (e.g. in which stores) I can find sustainable products. (R) | 2.80 (1.26) | | | | .59 |
| I can easily find sustainable products where I usually shop. (R) | 2.98 (1.18) | | | | .57 |
| I carefully plan my purchases in advance so that I can buy green products. (R) | 2.42 (1.24) | .33 | | | .47 |

(R) indicates which items were reverse-scored

# Means and standard deviations for each item were calculated prior to reverse-scoring, so as to avoid confusion in their interpretation.

Close to half of the sample reported never engaging in any action related to purchasing SPO, which is unsurprising and perhaps reflective of the nature of barriers that consumers face [45, 56]. This highlights the enormous potential to promote the purchasing of SPO in a behaviour change intervention. The lack of widespread knowledge about palm oil and the environmental and socio-economic issues that surround it emerged as a crucial barrier to consumer

**Table 3. Predicting purchasing of SPO products from the four COM-B factors of Pro-Green Consumption Attitudes, Knowledge, Demotivating Beliefs and Perceived Product Availability.**

| Predictors# | M (SD) | 95% CI for B | | | r | $sr^2$ |
|---|---|---|---|---|---|---|
| | | B | LB | UB | | |
| Fewer Pro-Green Consumption Attitudes | 2.51 (0.90) | -1.03* | -1.70 | -0.36 | -0.47** | 0.01 |
| Reduced Knowledge & Awareness | 3.16 (1.18) | -3.96** | -4.42 | -3.49 | -0.68** | 0.18 |
| De-motivating Beliefs | 2.83 (0.72) | 0.53 | -0.14 | 1.19 | -0.15** | 0.00 |
| Lower Perceived Product Availability | 3.27 (1.02) | -1.61** | -2.18 | -1.04 | -0.53** | 0.02 |

M = mean, SD = standard deviation, B = unstandardised beta, CI = confidence interval, LB = lower bound, UB = upper bound, r = correlation coefficient, $sr^2$ = squared semipartial correlation coefficient.

# All predictors have been scored as barriers

* $p < .01$

** $p < .001$, $R = .71$, $R^2 = .50$, $Adj R^2 = .50$.

PEB change. The importance of knowledge for PEB has been supported by previous studies [53, 92, 93], although when compared to other PEB like recycling or energy conservation which are relatively well-known, individuals may lack specific knowledge about the impact of their consumer choices on the environment [94]. While prior research has indicated that the more knowledge consumers had about palm oil and its environmental impacts, the greater the intentions to change their consumption habits [94], palm oil is a very complex issue, and so attaining relevant knowledge can be quite difficult and time-consuming [56]. Further, there are conflicting messages on purchasing SPO and boycotting palm oil, which could confuse consumers as to which source of information and advice is trustworthy. Consumers may also find it difficult to distinguish between companies that adopt sustainable or unethical/unsustainable practices [95, 96]. There is potential for future research to study how different levels of knowledge on palm oil (and its complexity) might relate to varying levels of engagement. However, consumers might engage more with the palm oil issue, if in addition to knowledge, clear, consistent, and directive information was provided and presented on product labels.

While clear, accurate and evidence-based knowledge is an essential capability to possess, that alone may be insufficient for purchasing SPO. The ease of availability of sustainable or green products is an important opportunity that facilitates green consumerism [48, 53, 55, 97]. Closely linked with product availability is visibility in places of shopping and clear labelling. Ingredients could be written in small letters, the sustainability logo used might be unfamiliar and manufacturers may not label palm oil on their products due to the negative public perceptions surrounding it [98–100]. These further make it difficult for consumers to obtain accurate information about product availability and hence may not perceive that SPO products are readily available.

With respect to motivation-related factors, the term "pro-green consumption attitudes" in this study subsumes feelings of environmental concern, empathy, responsibility, guilt, and perceived consumer efficacy, as well as pride and satisfaction at having performed consumer-related PEB. All these elements are supported by previous literature on green consumer behaviour [53, 54, 101–106]. However, traditionally, there has been a focus on placing the heavy responsibility of sustainable purchasing entirely on a consumer [107], effectively ignoring other factors (e.g., relevant capability and opportunity issues) that may be beyond a person's control [56, 91]. In this context, the current study addresses this by situating motivation alongside factors of the capability of possessing relevant and accurate information and the opportunity of perceived availability of products where people generally shop, rather than viewing it in isolation.

## Implications

The continuing fiery destruction of tropical rainforests is a pressing issue, having implications in climate change, biodiversity loss, and human health and life [8]. The current study is part of a series of step-by-step projects aimed at designing an effective intervention to assist with resolving the palm oil crisis. It is unique, as it is the first to apply the COM-B model and the BCW to green consumerism. Understanding barriers that go beyond consumer motivation and individual responsibility is essential to design effective and appropriate interventions [58, 108]. The results of this study highlight that any potential intervention to increase the purchase of SPO would not only need to increase concern for the environment, empathy, feelings of personal responsibility, and perceived consumer efficacy (relating to motivation), but more importantly address other barriers by: 1) providing adequate knowledge about the various aspects of the palm oil issue (enhancing capability); and 2) assisting consumers in finding SPO products (providing opportunity).

## Limitations

An unavoidable limitation of this study was that the Palm Oil-Related COM-B survey's items (except those on knowledge and awareness) pertained to green consumerism in general, rather than being SPO-specific. Given that knowledge and awareness about the palm oil issue is not widespread among the general public, using SPO-specific items for the other barriers might have resulted in invalid data, as participants likely would have found it difficult to answer the questions if they did not know what SPO was. However, as a result of this essential modification in the survey items, the scale measures barriers to green consumerism in general, which is assumed to be the umbrella term under which purchasing SPO would fall.

Finally, even though participants reported a range of frequencies of their past SPO-related consumer behaviour, it is important to note that most people were at the lower end of the range, and hence very few people actually reported a high level of engagement in SPO-related purchasing. This could imply that there is increased scope for interventions to potentially increase the purchasing of SPO products.

## Conclusion

We conducted a survey of 781 Australian consumers, and found that close to half the sample had never purchased products containing sustainable palm oil (SPO). Applying the Capability-Opportunity-Motivation model of understanding behaviour (COM-B), significant barriers to purchasing these sustainable products included a lack of knowledge about issues associated with palm oil production, uncertainty about product availability, and weak green consumption attitudes. These barriers significantly predicted low engagement in the target behaviour of purchasing products with SPO. As this study has successfully applied the COM-B model in consumerism, it lends support to the utility of applying this model and the larger Behaviour Change Wheel [58] in promoting other sustainable consumer behaviour. Finally, this study provides an important foundation for designing interventions to increase SPO consumer behaviour, as it highlights the specific internal and external barriers–beyond individual motivation–that need to be addressed.

## Supporting information

**S1 File. Complete palm oil-COM-B survey.**
(DOCX)

## Author Contributions

**Conceptualization:** Cassandra Shruti Sundaraja, Donald W. Hine, Amy D. Lykins.

**Data curation:** Cassandra Shruti Sundaraja.

**Formal analysis:** Cassandra Shruti Sundaraja.

**Investigation:** Cassandra Shruti Sundaraja.

**Methodology:** Cassandra Shruti Sundaraja, Donald W. Hine.

**Project administration:** Cassandra Shruti Sundaraja, Amy D. Lykins.

**Supervision:** Donald W. Hine, Amy D. Lykins.

**Writing – original draft:** Cassandra Shruti Sundaraja.

**Writing – review & editing:** Donald W. Hine, Amy D. Lykins.

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
