## [Decision Letter · Decision Letter 0]

22 Feb 2021

PONE-D-20-38505

Palm Oil: Understanding Barriers to Sustainable Consumption

PLOS ONE

Dear Dr.Amy Lykins

Thank you for submitting your manuscript to PLOS ONE. After careful consideration, we feel that it has merit but does not fully meet PLOS ONE’s publication criteria as it currently stands. Therefore, we invite you to submit a revised version of the manuscript that addresses the points raised during the review process.

We look forward to receiving your revised manuscript.

Kind regards,

Shalini Dhyani, Ph.D

Academic Editor

PLOS ONE

3. PLOS ONE has specific requirements for studies that are presenting a new method or tool as the primary focus, including a newly developed questionnaire or scale (https://journals.plos.org/plosone/s/submission-guidelines#loc-methods-software-databases-and-tools). One requirement is that the questionnaire or scale must be openly available under a license no more restrictive than CC BY. Accordingly, please include a copy of your questionnaire, scale, and/or interview guide as a Supporting Information file or provide a link if it is available through an online repository.

*In line with PLOS' guidelines on reproducibility (https://journals.plos.org/plosone/s/criteria-for-publication#loc-3), please also provide additional detail regarding how data extraction and analysis was performed on the qualitative component of your study, i.e., the interviews, including who conducted the interviews.

*Please note that peer review at PLOS ONE is not double-blinded (https://journals.plos.org/plosone/s/editorial-and-peer-review-process). For this reason, authors should include in the revised manuscript all the information removed for blind review, including the name of the institution(s) where the research was conducted.

Reviewers' comments:

Reviewer #1: The current study intends to generate information related to palm oil consumerism- behaviour and awareness in Australian consumers. It’s interesting to see that authors have generated information from 781 consumers through various methods. The authors also explore the possibility of capability-opportunity-motivation (COM-B) factors that influence consumer behaviour and further how the behaviour change wheel can be reorganised. The authors explored the concept of sustainability in terms of sustainable palm oil (SPO), however the manuscript misses to build upon the cause and effect relationship. The various components of capability-opportunity-motivation needs comprehensive explanations to its usage and how it can be incorporated in the palm oil sustainable consumption. The authors need to restructure the manuscript in terms of – pre-assessment, problem structuring, evaluation and further reflections based on the outcome. At each of this stage it should include aims, tools and methods, inference and what aspects of questionnaire included/answered. The concept of access before assess has to be explored. With regret, I must say that the manuscript is unacceptable for publication at the present stage.

Reviewer #2: This is an interesting study of the factors that determine whether and why Australians would buy sustainable palm oil products, with results indicating that that barriers related to a lack of knowledge and awareness about the issue reduced perceived product availability, and fewer pro-green consumption attitudes best predicted engagement in sustainable palm oil purchasing behaviour. I only have some comments about the text and requested further details about the methods, but overall found this an interesting and methodologically sound study. One minor issue is that the authors use abbreviations quite a bit. As PLOSONE does not have a word limit, I suggest writing them out in full throughout the text as it makes it much easier to read and the reader doesn’t need to go back to the start of the text to check the meaning of these abbreviations.

Page 8 of pdf (no page or line numbers were available). Abstract. The authors write that growing demand for palm oil had led to the burning of tropical forests. This is only partially correct. While land clearing for agriculture using fire has resulted in deforestation, especially in major recent fire years such as 2015 and 2019, the use of fire to open up land in industrial scale oil palm is now quite minimal and most fires relate to smallholder agricultural. That oil palm expansion has resulted in deforestation and biodiversity loss is undeniable but stating that fire is the key driver may have been correct in the 1990s but does not apply to the recent situations. For some relevant references, see here:

Gaveau DLA, Locatelli B, Descals A, Manurung T, Salim MA, Husnayen, Angelsen A, Meijaard E, Sheil D. 2021. Slowing oil palm expansion and deforestation in Indonesia coincide with low oil prices. Research Square. https://www.researchsquare.com/article/rs-143515/v1

Gaveau DLA, Pirard R, Salim MA, Tonoto P, Yaen H, Parks SA, Carmenta R. 2017. Overlapping Land Claims Limit the Use of Satellites to Monitor No-Deforestation Commitments and No-Burning Compliance. Conservation Letters 10:257-264.

Gaveau DLA, Sheil D, Husnayaen M, Salim A, Ancrenaz M, Pacheco P, Meijaard E. 2016. Rapid conversions and avoided deforestation: examining four decades of industrial plantation expansion in Borneo. Scientific Reports 6:32017.

Authors should spell out “COM-B” in the abstract because to most readers the term doesn’t mean anything.

Page 9. Introduction

The term “climate-related events” is quite vague. Lots of events are climate related. It would probably be better to refer to global warming or global heating or global climate change rather than generically to “climate”.

Again, the authors suggest that the burning of forests for agriculture in SE Asia is the main driver of deforestation. This is not quite accurate as mentioned above.

Page 10. Sun bear and Sumatran Elephant are Vulnerable and Endangered respectively on the IUCN Red List. Only the Bornean orangutan is Critically Endangered.

Page 11. RSPO is Roundtable on, not for Sustainable Palm Oil

Page 14. The authors write that the preliminary study results were subjected to framework analysis. Many readers might not know what this is, and some further methodological details on how the factors in Table 1 were obtained from the initial 13 interviews would be useful.

Page 15. The authors need to clarify the characteristics of the population from which their 781 sample was taken. Does the Qualtrics database only contain Australians, or also people from other countries. How do people end up in the database? It is important to know which population the sample represents. Also, did the participants volunteer to participate and if so does this introduce a certain bias (e.g., participants with limited interest in environmental issues may decline to participate). More information about this would be useful.

Page 16. What is PEB? Also, see my note above about avoiding abbreviations and writing them out in full.

Page 17. I suggest that the authors explain in a few words what the Cronbach’s alpha measures and what it means that it increased to 0.88 when one question was excluded from the analysis.

Page 19. The authors write “de Groot and Steg (68) adapted the original values scale to examine egoistic, altruistic and biospheric value orientations.” It is not clear to which original values this refers and authors should provide this context. Also the next sentence then mention 13 values, which presumably refer to the Schwartz’s value scale, but it is not explained what this scale is and what the values are. More explanation is needed here, so the reader can understand the methodological details.

Page 25. The finding that knowledge about palm oil is a very important determinant of consumer behaviour is interesting. As the authors point out, though, it is difficult for consumers to determine which information sources to trust, as there are many different ways to interpret knowledge about palm oil sustainability. For example, one scientific study showing that palm oil certification has a minimal positive impact on reducing deforestation will by some groups be interpreted as “certification is positive” and by others as “certification makes no difference to non-certification”. This is the nature of this polarized debate around palm oil and other vegetable oils. Do the authors have any insights as to how science could get better in informing this debate and getting more consumers to pay attention to relevant information, or understand why and how different interest groups might want to push certain views on palm oil sustainability?

Page 27. One of the issues to address for increasing the purchase of sustainable palm oil is to “provide adequate knowledge about various aspects of the palm oil issue”. How important is it, according to the authors, to also clarify the interchangeability of vegetable oils and the fact that all oil crops have impacts, as was, for example, suggested for coconut oil: Meijaard E, Abrams JF, Juffe-Bignoli D, Voigt M, Sheil D. 2020. Coconut oil, conservation and the conscientious consumer. Current Biology 30:R757-R758. Are consumers interested in these kinds of considerations, or do they want simple guidelines: buy this or do not buy that?

Page 27. In their text on limitations, the authors address my question above about the population from which their sample was drawn (Australian, mostly female). This should be described in the Methods.

Page 27. The authors write that “Given that knowledge and awareness about the palm oil issue is not widespread among the general public….”, but Table 2 indicates that the question “I am aware of the effects of palm oil production on forests” was a strong predictor. Without seeing the actual responses to the individual questions it is not possible for the reader to determine whether indeed the participants had limited knowledge about palm oil, and I suggest that the authors include a table (in the main text or as appendix) showing the average responses to each of the questions.

Reviewer's Responses to Questions

**Comments to the Author**

1. Is the manuscript technically sound, and do the data support the conclusions?

Reviewer #1: Partly

Reviewer #2: Yes

2. Has the statistical analysis been performed appropriately and rigorously? 

Reviewer #1: No

Reviewer #2: Yes

3. Have the authors made all data underlying the findings in their manuscript fully available?

Reviewer #1: Yes

Reviewer #2: No

4. Is the manuscript presented in an intelligible fashion and written in standard English?

Reviewer #1: No

Reviewer #2: Yes

---

## [Author Response · Author response to Decision Letter 0]

30 Mar 2021

Thank you to both reviewers for taking the time to read our paper and provide feedback. 

Reviewer #1’s comments: The current study intends to generate information related to palm oil consumerism- behaviour and awareness in Australian consumers. It’s interesting to see that authors have generated information from 781 consumers through various methods. The authors also explore the possibility of capability-opportunity-motivation (COM-B) factors that influence consumer behaviour and further how the behaviour change wheel can be reorganised. The authors explored the concept of sustainability in terms of sustainable palm oil (SPO), however the manuscript misses to build upon the cause and effect relationship. The various components of capability-opportunity-motivation needs comprehensive explanations to its usage and how it can be incorporated in the palm oil sustainable consumption. The authors need to restructure the manuscript in terms of – pre-assessment, problem structuring, evaluation and further reflections based on the outcome. At each of this stage it should include aims, tools and methods, inference and what aspects of questionnaire included/answered. The concept of access before assess has to be explored. With regret, I must say that the manuscript is unacceptable for publication at the present stage.

Response: Thank you for the feedback. We have incorporated more information on capability, opportunity and motivation, its usage and potential application in the context of sustainable palm oil (page 7, lines 149-163). However, in response to the other comments, we’re concerned that the reviewer may not have understood the purpose of our study. Our study was a quantitative study conducted in the form of an online survey at a single point in time. It was not our intention (nor would it be appropriate for us) to re-organize Michie et al.’s (2011) behaviour change wheel nor demonstrate true cause-effect relationships. We do not believe that the restructure this reviewer has suggested is appropriate given the aims and nature of our study. In our Materials and Methods (pages 10-16) section, we have provided information on the participants, measures used, procedure, data analysis and ethics. Further, we have included an additional paragraph (page 8-9; lines 186-192) in order to further clarify the purpose and scope of the current research. 

Reviewer #2’s comments: This is an interesting study of the factors that determine whether and why Australians would buy sustainable palm oil products, with results indicating that that barriers related to a lack of knowledge and awareness about the issue reduced perceived product availability, and fewer pro-green consumption attitudes best predicted engagement in sustainable palm oil purchasing behaviour. I only have some comments about the text and requested further details about the methods, but overall found this an interesting and methodologically sound study. One minor issue is that the authors use abbreviations quite a bit. As PLOSONE does not have a word limit, I suggest writing them out in full throughout the text as it makes it much easier to read and the reader doesn’t need to go back to the start of the text to check the meaning of these abbreviations. – 

Response: Thank you for this suggestion, We have removed all abbreviations, except the COM-B, which is commonly used by the developers of the behaviour change wheel (Michie et al., 2011), and in other research utilising the COM-B model. 

We have also inserted page numbers. Thank you for pointing out this omission. 

Abstract. The authors write that growing demand for palm oil had led to the burning of tropical forests. This is only partially correct. While land clearing for agriculture using fire has resulted in deforestation, especially in major recent fire years such as 2015 and 2019, the use of fire to open up land in industrial scale oil palm is now quite minimal and most fires relate to smallholder agricultural. That oil palm expansion has resulted in deforestation and biodiversity loss is undeniable but stating that fire is the key driver may have been correct in the 1990s but does not apply to the recent situations. For some relevant references, see here:

Gaveau DLA, Locatelli B, Descals A, Manurung T, Salim MA, Husnayen, Angelsen A, Meijaard E, Sheil D. 2021. Slowing oil palm expansion and deforestation in Indonesia coincide with low oil prices. Research Square. https://www.researchsquare.com/article/rs-143515/v1

Gaveau DLA, Pirard R, Salim MA, Tonoto P, Yaen H, Parks SA, Carmenta R. 2017. Overlapping Land Claims Limit the Use of Satellites to Monitor No-Deforestation Commitments and No-Burning Compliance. Conservation Letters 10:257-264.

Gaveau DLA, Sheil D, Husnayaen M, Salim A, Ancrenaz M, Pacheco P, Meijaard E. 2016. Rapid conversions and avoided deforestation: examining four decades of industrial plantation expansion in Borneo. Scientific Reports 6:32017.

Response: Thank you for providing this information and useful clarification regarding the role of forest fires. We have reviewed the suggested articles and have made appropriate amendments in the abstract and introduction (page 2, lines 22-23; page 4, lines 80, 85-87). 

Authors should spell out “COM-B” in the abstract because to most readers the term doesn’t mean anything.

Response: Thank you for this suggestion. We have incorporated the expansion of COM-B in the abstract (page 2, line 25). 

Page 9. Introduction

The term “climate-related events” is quite vague. Lots of events are climate related. It would probably be better to refer to global warming or global heating or global climate change rather than generically to “climate”.

Response: Thank you for this suggestion. We have changed it to ‘climate change’ (page 3, line 44). 

Again, the authors suggest that the burning of forests for agriculture in SE Asia is the main driver of deforestation. This is not quite accurate as mentioned above.

Response: Thank you. This has been amended as described above.

Page 10. Sun bear and Sumatran Elephant are Vulnerable and Endangered respectively on the IUCN Red List. Only the Bornean orangutan is Critically Endangered.

Response: Thank you for pointing out this error. This has been corrected accordingly (page 4. line 79). 

Page 11. RSPO is Roundtable on, not for Sustainable Palm Oil

Response: Thank you for also pointing out this error. This has been corrected accordingly (page 5, line 98). 

Page 14. The authors write that the preliminary study results were subjected to framework analysis. Many readers might not know what this is, and some further methodological details on how the factors in Table 1 were obtained from the initial 13 interviews would be useful. – Response: Thank you for this feedback. We have included a brief explanation on framework analysis (page 9, lines 207-212). 

Page 15. The authors need to clarify the characteristics of the population from which their 781 sample was taken. Does the Qualtrics database only contain Australians, or also people from other countries. How do people end up in the database? It is important to know which population the sample represents. Also, did the participants volunteer to participate and if so does this introduce a certain bias (e.g., participants with limited interest in environmental issues may decline to participate). More information about this would be useful.

Page 27. In their text on limitations, the authors address my question above about the population from which their sample was drawn (Australian, mostly female). This should be described in the Methods.

Response: Thank you for these suggestions. We have added to the existing demographic information by including more information on the distribution across age groups. We have also elaborated on the Qualtrics invitation process and how they source samples (pages 10-11, lines 231-237).

On reflection and discussion, we agreed that the non-representative sample (of mostly female participants) was not a limitation of this study as women are more likely to do the grocery shopping for the household. The information under ‘Limitations’ has been re-worded and placed under ‘Participants’ (page 11, lines 238-243). 

Page 16. What is PEB? Also, see my note above about avoiding abbreviations and writing them out in full.

Response: Thank you. As mentioned above, we have avoided all abbreviations (except for COM-B) for the aforementioned reasons. 

Page 17. I suggest that the authors explain in a few words what the Cronbach’s alpha measures and what it means that it increased to 0.88 when one question was excluded from the analysis. 

Response: Thank you for this suggestion. We have included this (page 12-13, lines 272-277). 

Page 19. The authors write “de Groot and Steg (68) adapted the original values scale to examine egoistic, altruistic and biospheric value orientations.” It is not clear to which original values this refers and authors should provide this context. Also the next sentence then mention 13 values, which presumably refer to the Schwartz’s value scale, but it is not explained what this scale is and what the values are. More explanation is needed here, so the reader can understand the methodological details.

Response: Thank you for pointing this out. This paragraph has been amended with a bit more information on de Groot & Steg’s version of Schwartz’s original values (page 14-15, lines 317-320).

Page 25. The finding that knowledge about palm oil is a very important determinant of consumer behaviour is interesting. As the authors point out, though, it is difficult for consumers to determine which information sources to trust, as there are many different ways to interpret knowledge about palm oil sustainability. For example, one scientific study showing that palm oil certification has a minimal positive impact on reducing deforestation will by some groups be interpreted as “certification is positive” and by others as “certification makes no difference to non-certification”. This is the nature of this polarized debate around palm oil and other vegetable oils. Do the authors have any insights as to how science could get better in informing this debate and getting more consumers to pay attention to relevant information, or understand why and how different interest groups might want to push certain views on palm oil sustainability?

Response: Thank you for raising the question on what kind of information would serve to best engage consumers. We have included this in our ‘Discussion’ as scope for potential future research (page 22, lines 436-440). With respect to the second question, we agree that there could be different interest groups that might want to push certain views on palm oil sustainability, and this has been touched upon in a previous research where we interviewed experts from various fields in attempting to understand the possible actions that consumers could engage in and the reasons behind them (Sundaraja et al. 2020).

Page 27. One of the issues to address for increasing the purchase of sustainable palm oil is to “provide adequate knowledge about various aspects of the palm oil issue”. How important is it, according to the authors, to also clarify the interchangeability of vegetable oils and the fact that all oil crops have impacts, as was, for example, suggested for coconut oil: Meijaard E, Abrams JF, Juffe-Bignoli D, Voigt M, Sheil D. 2020. Coconut oil, conservation and the conscientious consumer. Current Biology 30:R757-R758. Are consumers interested in these kinds of considerations, or do they want simple guidelines: buy this or do not buy that?

Response: Thank you for raising this other important point. Research has been clear that palm oil has a higher yield than any other vegetable oil (Meijaard et al., 2018), and we have now explicitly added this to the Introduction (page 4; lines 88-92). As mentioned above, future research will have to determine how much information consumers actually need to encourage them towards sustainable palm oil purchases (page 22, lines 436-440). The kinds of considerations that consumers might better respond to is perhaps beyond the scope of this current paper and is better covered in our manuscript on the qualitative framework analysis. 

Page 27. The authors write that “Given that knowledge and awareness about the palm oil issue is not widespread among the general public….”, but Table 2 indicates that the question “I am aware of the effects of palm oil production on forests” was a strong predictor. 

Response: Thank you. We would however like to clarify that the numerical values in Table 2 indicated a loading on a factor (the correlation between the item and the factor) and were not indicative of the strength of the association between item and SPO purchasing behaviour. 

Without seeing the actual responses to the individual questions it is not possible for the reader to determine whether indeed the participants had limited knowledge about palm oil, and I suggest that the authors include a table (in the main text or as appendix) showing the average responses to each of the questions. 

Response: Thank you for this suggestion. We have included an additional column in Table 2 that provides the means and standard deviation for each item, and an additional column in Table 3, indicating means and standard deviations of the factor scores. We have also indicated when the item was reverse scored, as all items in Table 2 were interpreted as barriers (pages 18-19).

---

## [Decision Letter · Decision Letter 1]

3 Jun 2021

PONE-D-20-38505R1

Palm Oil: Understanding Barriers to Sustainable Consumption

PLOS ONE

Dear Dr. Lykins,

Thank you for submitting your manuscript to PLOS ONE. After careful consideration, we feel that it has merit but does not fully meet PLOS ONE’s publication criteria as it currently stands. Therefore, we invite you to submit a revised version of the manuscript that addresses the points raised during the review process.

We look forward to receiving your revised manuscript.

Kind regards,

Ali B. Mahmoud, Ph.D.

Academic Editor

PLOS ONE

Reviewers' comments:

Reviewer's Responses to Questions

**Comments to the Author**

1. If the authors have adequately addressed your comments raised in a previous round of review and you feel that this manuscript is now acceptable for publication, you may indicate that here to bypass the “Comments to the Author” section, enter your conflict of interest statement in the “Confidential to Editor” section, and submit your "Accept" recommendation.

Reviewer #2: (No Response)

Reviewer #3: All comments have been addressed

2. Is the manuscript technically sound, and do the data support the conclusions?

Reviewer #2: Yes

Reviewer #3: Yes

3. Has the statistical analysis been performed appropriately and rigorously? 

Reviewer #2: Yes

Reviewer #3: Yes

4. Have the authors made all data underlying the findings in their manuscript fully available?

Reviewer #2: Yes

Reviewer #3: Yes

5. Is the manuscript presented in an intelligible fashion and written in standard English?

Reviewer #2: Yes

Reviewer #3: Yes

6. Review Comments to the Author

Reviewer #2: I reiterate that this is an interesting study on green consumerism with a view on the use of sustainable palm oil in Australia. It uses a novel methodological approach for this field and provide useful insights, at least for the Australian context (I guess that repeating the questions in poor parts of Indonesia or India would result in very different outcomes).

The authors have addressed my original comments adequately and I only have minor suggestions now.

Thank you for writing out most of the abbreviations in the text. I still think that using the abbreviation "COM-B" in the abstract is confusing, because many people won't know what it means and abstracts should really be crystal clear to the lay reader. COM-B is an element of BCW and not well known outside specific behavioural change studies.

By the way, there are still no page or line numbers which makes it more difficult to comment on the text. There are some minor grammatical and spelling issues and I suggest that the authors proofread their text carefully before final submission.

Line 2 in Abstract: I suggest just writing: "expansion of oil palm plantations has led to the loss of tropical forests and biodiversity loss. Fire just isn't the biggest factor in deforestation.

Reviewer #3: This resubmitted paper addresses an extremely important issue. When reading it, it becomes clear that the authors have already worked in detail on the present version. I therefore have only a few comments that should help to further improve the paper. As someone using factor analysis quite often, I really enjoyed reading that you used an oblimin rotation. Usually, I al-ways read varimax.

I would very much welcome it if the use of palm oil were briefly explained in more detail. Please point out that it is not only used in food and animal feed but also, for example, in cosmetics, detergents and also as fuel.

The explanation on the preliminary study is good, but should already be written where the preliminary study is mentioned for the first time.

Table 3: Please explain all abbreviations. What is B, UB, LB etc.? Most of the readers will prob-ably know it but not everyone.

I suggest that section “Participants section” and “Procedure” be placed directly one after the other.

The discussion is very short and the comparison with the current literature is very poor. Please include more recent literature. A lot has been published on the topic of consumer preferences and palm oil in the last few years.

7. PLOS authors have the option to publish the peer review history of their article (what does this mean?). If published, this will include your full peer review and any attached files.

Reviewer #2: No

Reviewer #3: No

---

## [Author Response · Author response to Decision Letter 1]

23 Jun 2021

Response to Reviewer #1

Thank you for writing out most of the abbreviations in the text. I still think that using the abbreviation "COM-B" in the abstract is confusing, because many people won't know what it means and abstracts should really be crystal clear to the lay reader. COM-B is an element of BCW and not well known outside specific behavioural change studies.

Thank you for this comment. The acronym COM-B is expanded upon in lines 23-24 of the abstract. 

“The current study is the first to employ a capability-opportunity-motivation (COM-B) framework in green consumerism, to determine which capability, opportunity, and motivation factors strongly predict the intentional purchasing of sustainable palm oil products by Australian consumers (N = 781).”

By the way, there are still no page or line numbers which makes it more difficult to comment on the text. There are some minor grammatical and spelling issues and I suggest that the authors proofread their text carefully before final submission.

Page numbers and line numbers are included. The manuscript has been proof-read once again – thank you. 

Line 2 in Abstract: I suggest just writing: "expansion of oil palm plantations has led to the loss of tropical forests and biodiversity loss. Fire just isn't the biggest factor in deforestation.

Thank you – this line had been changed in the revised manuscript that was submitted, with the reference to fire removed (lines 22-23):

“However, the growing demand for palm oil is leading to deforestation and biodiversity loss.”

Response to Reviewer #2

I would very much welcome it if the use of palm oil were briefly explained in more detail. Please point out that it is not only used in food and animal feed but also, for example, in cosmetics, detergents and also as fuel.

Thank you for this suggestion. We have included lines 71-77 including information on the use of palm oil.

The explanation on the preliminary study is good, but should already be written where the preliminary study is mentioned for the first time.

Thank you for this suggestion. This entire section has been re-written as Study 1, on the request of the editor. 

I suggest that section “Participants section” and “Procedure” be placed directly one after the other.

Thank you for this suggestion. However, we believe that the current order of “Participants”, “Measures” and then “Procedure” is suited to describe this research study, as the measures used and described under “Measures” are subsequently referred to under “Procedure”. Changing this order might result in losing the coherent flow between these sections. 

Table 3: Please explain all abbreviations. What is B, UB, LB etc.? Most of the readers will prob-ably know it but not everyone.

Thank you for this suggestion. We have now included a note under the table, expanding on all the abbreviations used (lines 425-429, page 21).

The discussion is very short and the comparison with the current literature is very poor. Please include more recent literature. A lot has been published on the topic of consumer preferences and palm oil in the last few years.

Thank you for pointing this out. The discussion has now been expanded upon and updated with more recent literature (References 91, 95, 97, 100, 101).

---

## [Decision Letter · Decision Letter 2]

7 Jul 2021

Palm Oil: Understanding Barriers to Sustainable Consumption

PONE-D-20-38505R2

Dear Dr. Lykins,

We’re pleased to inform you that your manuscript has been judged scientifically suitable for publication and will be formally accepted for publication once it meets all outstanding technical requirements.

Kind regards,

Ali B. Mahmoud, Ph.D.

Academic Editor

PLOS ONE

Additional Editor Comments (optional):

Reviewers' comments:

Reviewer's Responses to Questions

**Comments to the Author**

1. If the authors have adequately addressed your comments raised in a previous round of review and you feel that this manuscript is now acceptable for publication, you may indicate that here to bypass the “Comments to the Author” section, enter your conflict of interest statement in the “Confidential to Editor” section, and submit your "Accept" recommendation.

Reviewer #2: All comments have been addressed

Reviewer #3: All comments have been addressed

2. Is the manuscript technically sound, and do the data support the conclusions?

Reviewer #2: Yes

Reviewer #3: Yes

3. Has the statistical analysis been performed appropriately and rigorously? 

Reviewer #2: Yes

Reviewer #3: Yes

4. Have the authors made all data underlying the findings in their manuscript fully available?

Reviewer #2: Yes

Reviewer #3: Yes

5. Is the manuscript presented in an intelligible fashion and written in standard English?

Reviewer #2: Yes

Reviewer #3: Yes

6. Review Comments to the Author

Reviewer #2: I am happy with the final minor edits the authors have made, including minor changes to the abstract and main text.

Reviewer #3: By revising this paper again, it has improved significantly. I see no reason why it should not be published now.

7. PLOS authors have the option to publish the peer review history of their article (what does this mean?). If published, this will include your full peer review and any attached files.

Reviewer #2: No

Reviewer #3: No

---

## [Editor Report · Acceptance letter]

19 Jul 2021

PONE-D-20-38505R2 

Palm Oil: Understanding Barriers to Sustainable Consumption 

Dear Dr. Lykins:

I'm pleased to inform you that your manuscript has been deemed suitable for publication in PLOS ONE. Congratulations! Your manuscript is now with our production department. 

Kind regards, 

on behalf of

Dr. Ali B. Mahmoud 

Academic Editor

PLOS ONE